# Hybrid Beamforming and Relay Selection for End-to-End SNR Maximization in Single-User Multi-Relay MIMO Systems

**DOI:** 10.3390/s23042079

**Published:** 2023-02-13

**Authors:** Hafiz Muhammad Tahir Mustafa, Jung-In Baik, Young-Hwan You, Hyoung-Kyu Song, Zunira Abbasi

**Affiliations:** 1Department of Information and Communication Engineering, Sejong University, Seoul 05006, Republic of Korea; 2Department of Convergence Engineering for Intelligent Drone, Sejong University, Seoul 05006, Republic of Korea; 3Department of Computer Engineering, Sejong University, Seoul 05006, Republic of Korea

**Keywords:** hybrid beamforming, millimeter wave, MIMO, non-regenerative relay

## Abstract

This paper proposes a novel hybrid beamforming and relay selection scheme for spectral efficiency maximization in a non-regenerative multi-relay multi-input multi-output (MIMO) system. The analog beamforming component in the radio-frequency (RF) domain must follow an element-wise constant modulus constraint, which makes the underlying design problem mathematically intractable and therefore, it is quite challenging to obtain the global optimal solution. To address this problem, phase-only precoding/combining matrices are derived by maximizing the end-to-end received signal-to-noise ratio (SNR) under transmit power constraint at the source and each relay node. This task is achieved by decomposing the original complicated optimization problem into two independent components. The first component designs the RF precoder/combiner at source and relay nodes by maximizing the received SNR at relay nodes. While the second component attempts to derive the analog precoder/combiner at relay nodes and destination by maximizing the received SNR at the destination. Digital baseband processing matrices are obtained by deriving the closed-form expression, which minimizes interference among different sub-channels. Finally, the relay selection is made by maximizing the overall SNR from the source to the destination. Computer simulations reveal that the performance of the proposed algorithm is close to its fully digital counterpart and approximately 6% higher than the specified relay-assisted hybrid beamforming techniques. Moreover, the proposed method achieves more than 15% higher performance in a sparse scattering environment when compared with the given relay selection techniques.

## 1. Introduction

The exponential growth of data rates to accommodate the rapid development of emerging data-hungry applications such as artificial intelligence, virtual reality, augmented reality, and holographic imaging have led to the exploration of underutilized mm-Wave frequency spectrum (30–300 GHz) for designing the future mobile wireless networks [1,2,3,4]. The transmitted signal at mm-Wave frequency suffers from huge path-loss and other channel impairments, and it poses a great challenge to establish a reliable non-line-of-sight (NLOS) communication link as mm-Wave signals are sensitive to blockage [1,4]. In addition, the sparsity of mm-Wave scattering environment usually leads to rank-deficient channels [5]. By taking the advantage of short wavelength at mm-Wave frequencies, the poor characteristics associated with mm-Wave transmission (e.g., severe path loss, atmospheric absorptions, high penetration loss etc.) can be depressed by deploying the large number of antennas in a small region to achieve significant beamforming gain for controlling the interference among different users/cells [6].

Multi-hop communication is an essential technology which reduces the dependency on the network infrastructure and this paradigm makes it possible for multiple satellites or multiple un-manned aerial vehicles (UAVs) to communicate [7,8]. In addition, multi-hop communication provides assistance in controlling the problem of deep fading over long-distance transmission, when mm-Wave frequency band or Terahertz frequency spectrum is employed for signal transmission [9,10]. The notion of relay-assisted communication is incorporated in MIMO systems to enhance network coverage and to make long-distance transmission reliable by improving the link quality. Regenerative and non-regenerative operations are two main signal processing strategies at relay nodes. In regenerative schemes, the received signal is decoded first at each intermediate hop before transmitting to the next hop, while non-regenerative schemes forward the received signal after amplification. Low complexity and high security are the characteristics of non-regenerative techniques [11].

The authors of [12] proposed the beamforming algorithm for amplify-and-forward (AF) relay networks by using statistical channel state information at the transmitter. While the authors of [13] presented a distributed beamforming method which attempts to maximize the received SNR. This technique requires full instantaneous channel state information (SCI) at communicating nodes to determine power allocation. It is of great practical importance to investigate the performance of hybrid beamforming design for mm-Wave MIMO relay systems. In general, this paradigm increases the probability to have LOS communication between source and destination indirectly [14,15].

Fully digital massive MIMO systems have tremendous advantages in providing high spectral efficiencies. However, equipping each antenna with a dedicated RF chain significantly increases the cost and power consumption of such systems. Hence, hybrid beamformers have attracted a lot of attention as a means to provide a trade-off between spectral and energy efficiencies, [16,17]. In such systems, a small number of RF chains are connected to a large number of antennas via a network of phase shifters and/or switches.

Fully digital precoding for mm-Wave massive MIMO systems is not feasible in actual practice as it requires one dedicated RF chain for each antenna element in an array [18]. The RF chain contains analog-to-digital/digital-to-analog converter (ADC/DAC), amplifiers, mixers, and low-noise amplifiers, and therefore it is quite expensive and power hungry as well. In turn, it increases hardware complexity, cost, and power consumption to the system [19]. To overcome this problem, the notion of hybrid beamforming was introduced to achieve a good compromise between complexity and performance, where full complexity precoding is decomposed into the analog component for beamforming gain and a low-dimensional digital baseband processing unit to obtain the multiplexing gain [18,19,20]. A network of phase shifters/switches is required for practical implementation of the analog RF processor as it follows an element-wise constant amplitude constraint. However, there is no restriction on digital baseband precoder/combiner except power constraint. Usually, two types of hybrid precoding architectures are commonly used: (1) fully-connected architecture in which each RF chain is connected to all antenna elements in an array, and (2) partially-connected architecture in which each RF chain is connected to a sub-set of antennas [21]. It is worthwhile to mention here that fully-connected architecture is one of the most popular implementations for hybrid beamforming due to its great potential in achieving performance close to the fully-digital counterpart.

In [21] and [22], the authors proposed hybrid precoding designs based on Euclidean distance minimization between the hybrid beamforming solution and the optimal beamformer. The authors of [23,24,25] proposed codebook-based hybrid transceiver designs, where the analog beamformers were selected from the pre-defined candidate vectors. The common examples of these pre-arranged candidates are beam-steering vectors and discrete Fourier transform (DFT) beamformers. Hybrid beamforming algorithms in [26,27,28] are based on the minimum mean squared error (MMSE) criterion for deriving the hybrid beamforming matrices.

In contrast to the fully-connected structure, the authors of [29,30,31] proposed partially-connected architectures for hybrid beamforming which is done to obtain the energy efficient solution with relatively low hardware complexity. This structure allows the connection of each RF chain to a fixed non-overlapping subset of the antennas while deriving the analog RF beamformer corresponding to this antenna sub-array. In particular, hybrid beamforming schemes with partially-connected architecture may not always achieve satisfactory performance [32,33] for the following reasons: (1) it is difficult to have fine control over the beam and this tendency leads to the less accurate beamforming; (2) the underlying structure limits the flexibility of large antenna arrays and causes significant degradation in performance.

In [34], orthogonal matching pursuit (OMP) based solution proposed in [23] was extended to design the relay-assisted MIMO network with AF relaying protocol. The hybrid transceiver design proposed in [23] was further modified in [35] and [36] to design multi-relay MIMO systems. In [35], only the analog precoders/combiners were employed at relay nodes, while hybrid processing components were used at source and destination. In contrast to the hybrid precoding design in [35], the proposed algorithm in [36] used hybrid beamformers at all communicating nodes. In [37], the digital baseband processing component and the analog RF beamformer were designed separately for a single relay multi-user MIMO system by using pre-defined codebooks, and this approach was adopted to avoid intractable searching problem. The authors of [38] proposed the hybrid beamforming scheme as a matrix factorization problem, where the alternating direction method of multipliers (ADMM) was exploited to design the analog RF and digital baseband processing components. This method does not require any pre-defined codebook for the selection of best beamforming vectors. The authors of [39] introduced the notion of deep neural network for training the hybrid precoder and combiner; the hybrid precoding solution achieved relatively superior performance when compared with the existing hybrid beamforming algorithms. Deep learning framework shifts the run-time computational complexity for designing the hybrid beamformers to the off-line training process.

Contrary to the aforementioned works on relay-assisted hybrid beamforming for different communication networks (e.g., [34,35,36,37,38]), this paper proposes a beamforming and relay selection simultaneously. Relay selection in a multi-hop transmission system is an essential design aspect to avoid performance degradation due to error propagation [40,41]. Therefore, the primary focus of present research work is to design a hybrid beamforming solution for a fully-connected multi-relay mm-Wave MIMO network with a relay selection that may enable to maximize the achievable rate. This target can be achieved by maximizing end-to-end SNR, which is a better choice as it is directly related to capacity expression [42]. In this regard, the original complicated optimization problem is decomposed into two independent SNR maximization problems. Then, a decoupled approach is considered which allows to design the analog RF precoders/combiners and the corresponding baseband processing components separately.

In the context of recent research on relay-assisted mm-Wave MIMO systems with hybrid beamforming architecture, one of the major design goals is to enhance the reliability of transmitted data, even under the worst channel impairments. This objective can be achieved by deploying multiple relays in the communication path between source and end-user. Moreover, this paradigm enables having several channels for data transmission from the source to the destination. It is highly unlikely that all the available links are suffering due to severe channel impairments. To enhance the spectral efficiency and overall system performance, some suitable mechanism is required for relay selection along with the hybrid beamforming design at all communicating nodes. Therefore, the proposed algorithm is an attempt to address hybrid transceiver design and relay selection while considering relay-based mm-Wave MIMO system. In contrast to an iterative approach and codebook-based method for hybrid precoding design, the proposed methodology is free from the iterative procedure and searching algorithm associated with codebook in finding the best possible solution. Furthermore, an analytic solution is provided for deriving the analog RF beamformers and digital baseband processing components at all communicating nodes. While relay selection is made by considering the highest SNR from the source to the destination. The contributions of the proposed scheme are summarized as follows:The proposed research work introduces the hybrid beamforming and relay selection mechanism in a single-user multi-relay mm-Wave MIMO system. In order to avoid performance degradation in a multi-hop transmission network due to poor channel characteristics, the proposed algorithm selects the relay node which gives maximum SNR from the source to the destination. This selection criterion enables maximizing the spectral efficiency even in the presence of unfavorable conditions on several transmission paths.The proposed hybrid beamforming scheme derives the common analog RF precoder at source by finding the orthonormal basis for column space spanned by all the channels from source to relay nodes. The common analog RF combiner at the destination is obtained by minimizing the error over all optimal combiners corresponding to the channels from relay nodes to the destination. Finally, the analog precoder/combiner at relay nodes are derived by solving the respective SNR maximization problems.By considering the equivalence of spectral efficiency maximization and MSE minimization, digital baseband processing components at communicating nodes are obtained under MMSE criterion. This design procedure leads to the condition for minimizing interference and hence, maximizing the spectral efficiency.Computer simulations were conducted to evaluate the performance of the proposed hybrid beamforming algorithm by varying system configuration parameters. It is obvious from the obtained results that the proposed approach has a great potential to show good performance in sparse and rich scattering environments such as mm-Wave and Rayleigh fading channels, respectively.The existing works (e.g., [34,35,36,37,38]) only focus on the hybrid beamforming design for relay-assisted mm-Wave MIMO systems under different scenarios. With this perspective, there is a need to address relay-assisted MIMO communication in conjunction with a relay selection mechanism to enhance spectral efficiency of the system. Therefore, the proposed methodology may be considered as an important application area in the scope of hybrid relaying, which is capable of avoiding performance degradation.

The rest of the paper is organized as follows. System model and problem formulation are given in Section 2. Analog RF beamforming design at different communicating nodes and relay selection scheme are presented in Section 3. Derivation of digital baseband processing units and complexity analysis are included in Section 4. Computer simulations to evaluate the performance of the proposed algorithm are provided in Section 5. Concluding remarks are given in Section 6.

Notation: Upper-case and lower-case letters A and a denote a matrix and vector, respectively. For a given matrix A;$\;{\left\|A\right\|}_{F}$, A(:,i), A(:,1:j), tr(A), $A^{H}$, ∡A and |A(i,j)| denote the Frobenius norm, *i*th column, first j columns, trace, transpose, conjugate transpose, element-wise phase, and element-wise modulus of matrix A, respectively. ℂ denotes the field of complex numbers and $I_{m}$ is the m×m identity matrix, and $CN\left(0,\;\sigma^{2}I_{n}\right)$ is the complex Gaussian distribution with mean 0 and covariance matrix $\sigma^{2}I_{n}$. ${⌴}_{r},\times_{r}\left(r=1,\;2,\;3\right)\;$represent the concatenation of tensors or matrices along the *r*th dimension [43] and r-mode product between a tensor and matrix [44], respectively.

## 2. System Model and Problem Formulation

The proposed hybrid transceiver for a two-hop multi-relay mm-Wave MIMO system with relay selection is shown in Figure 1, where Nt, Nrel and Nr are the number of antennas at source, relay, and destination, respectively. Let sk∈ℂNs×1 be a complex information symbol vector consisting of Ns transmitted data streams from the source to the destination through the *k*th selected relay from a set of K relay nodes, which is given as
(1)sk={s1,s2,…,sNs}T, E{skskH}=INs. 

The number of RF-chains at the source, relay node and destination are represented as$\;N_{t}^{RF},\;N_{rel}^{RF}$ and$\;N_{r}^{RF}$, respectively, and the essential condition that needs to be satisfied for reliable transmission of data streams, when hybrid precoding is taken into account, can be expressed as
(2)$$N_{s}\leq min\left(N_{t}^{RF},\;N_{rel}^{RF},\;N_{r}^{RF}\right)\ll min\left(N_{t},\;N_{rel},\;N_{r}\right).\;$$

The condition (2) clearly indicates that the number of RF chains determines the maximum number of data streams that can be transmitted reliably, and the number of RF chains is far less than the number of antennas which describes the potential of hybrid beamforming in making the system feasible on practical grounds.

It is assumed that a direct link between the source and destination is not feasible due to excessive path-loss and deep fading, which is typical of mm-Wave propagation. Furthermore, each relay node follows AF relaying protocol and all communicating nodes are supposed to be in half-duplex mode. In relay-assisted MIMO systems, the signal transmission can be divided into two time-slots. In the first time-slot, source transmits Ns data streams to all the relay nodes. In the second time-slot, the selected *k*th relay where k∈{1,2,…,K} re-transmits the signal to the end user. The transmitted signal from the source to the *k*th relay, after applying hybrid precoding, can be expressed as
(3)xk=VRFVBB,ksk ∈ℂNt×1, 
where VRF∈ℂNt×NtRF is the common analog RF beamformer and VBB,k∈ℂNtRF×Ns is the digital baseband precoding matrix corresponding to the *k*th selected relay. The received signal at the *k*th relay node is given as
(4)yrk=Hk VRFVBB,ksk+zr ∈ℂNrel×1, 
where Hk∈ℂNrel×Nt, zr∈ℂNrel×1 are the channel matrix between the source and the *k*th relay node and, zero mean circularly symmetric complex Gaussian (ZMCSCG) noise vector with variance σr2 i.e., zr~CN(0, σr2INrel), respectively.

Before proceeding further, it is worth mentioning that the hybrid beamforming structure at the relay node consists of two analog RF and the corresponding digital baseband processing components. One analog RF processing matrix acts as a receive beamformer for the source transmitted signal in the first time-slot. The other analog RF processing matrix is responsible for the transmit beamforming, which enables the transmission of the processed signal at the relay node to the destination in the second time-slot. Let F1k∈ℂNrel×NrelRF be the analog RF combiner and F2k∈ℂNrel×NrelRF be the analog precoder at the *k*th relay node. Similarly, the corresponding digital baseband combiner and precoder are denoted by GBB1,k∈ℂNrelRF×Ns and GBB2,k∈ℂNrelRF×Ns, respectively. These analog and digital processing units (F1k, F2k, GBB1,k, GBB1,k) constitute the hybrid filter Fk∈ℂNrel×Nrel at the *k*th relay node. The mathematical formulation of Fk∈ℂNrel×Nrel is written as
(5)Fk=F2kGBB2,kGBB1,kHF1kH=F2kGBB,kF1kH, 
where GBB,k=GBB2,kGBB1,kH∈ℂNrelRF×NrelRF is the overall digital baseband processing matrix at the *k*th relay node. When receive processing is applied on (4) then, the processed signal can be characterized as
(6)y1,k=GBB1,kHF1kHHk VRFVBB,ksk+GBB1,kHF1kHzr∈ℂNs×1. 

While designing a hybrid precoder at the source and a hybrid combiner at the relay node, it is desired to minimize interference among transmitted data streams from the source to relay node, which in turn maximizes SNR at the relay node. This task can be achieved by transforming the composite channel Hcomp1=GBB1,kHF1kHHk VRFVBB,k∈ℂNs×Ns into an equivalent parallel single-input single-output (SISO) channels. Using this formation, the received signal at relay corresponding to the *n*th data stream sn,k in sk (1) can be represented with the following input output relationship
(7)y1,kn=αn,k sn,k+v1,k, n∈{1,2,…,Ns}, 
where αn,k≜‖GBB1,kH(:,n)F1kHHk VRFVBB,k(:,n)‖ and the operator ‖.‖ represent the absolute value of a complex number and, v1,k~CN(0, σr2). Therefore, (7) can be expressed as
(8)y1,kn=‖GBB1,kH(:,n)F1kHHk VRFVBB,k(:,n)‖ sn,k+v1,k. 

Relay node performs hybrid beamforming on y1,k (6) for further transmission of this signal to the destination. The received signal at the destination can be written as
(9)y2,k~=GkF2kGBB2,ky1,k+zd∈ℂNr×1. 

Finally, the received signal after applying hybrid combiner at the destination can be modeled as
(10)y2,k=WBB,kHWRFH(GkF2kGBB2,ky1,k+zd)∈ℂNs×1, 
where WRF∈ℂNr×NrRF, WBB,k∈ℂNrRF×Ns, Gk∈ℂNr×Nrel, and zd∈ℂNr×1 are the common analog RF combiner, digital baseband combiner corresponding to the *k*th selected relay, channel matrix between the *k*th relay node and the final destination and ZMCSCG noise vector with variance σd2 i.e., zd~CN(0, σd2INr), respectively.

Similarly, it is also required to minimize interference, while designing a hybrid beamforming matrix at the relay node and destination, for efficient transmission of data streams from the selected relay node to the end user. This target can be achieved by converting the composite channel Hcomp2=WBB,kHWRFHGk F2kGBB2,k∈ℂNs×Ns into an equivalent parallel SISO sub-channels, and it facilitates in achieving SNR maximization at the destination. Therefore, the final received signal at the destination corresponding to the *n*th data stream sn,k in sk (1) can be expressed as
(11)y2,kn=ρn,k y1,kn+v2,k, n∈{1,2,…,Ns}, 
where ρn,k≜‖WBB,kH(:,n)WRFHGk F2kGBB2,k(:,n)‖ and v2,k~CN(0, σd2). Using (8), (11) can also be written as
(12)y2,kn=‖WBB,kH(:,n)WRFHGk F2kGBB2,k(:,n)‖ (‖GBB1,kH(:,n)F1kHHk VRFVBB,k(:,n)‖sn,k+v1,k)+v2,k. 

Therefore, the end-to-end SNR corresponding to $N_{s}$ data streams through the *k*th relay node can be expressed as
(13)$$\beta_{k}=\frac{\|W_{BB,k}^{H}W_{RF}^{H}G_{k}\;F_{2k}G_{BB2,k}{\|}_{F}^{2}\|G_{BB1,k}^{H}F_{1k}^{H}H_{k}\;V_{RF}V_{BB,k}{\|}_{F}^{2}}{\|W_{BB,k}^{H}W_{RF}^{H}G_{k}\;F_{2k}G_{BB2,k}{\|}_{F}^{2}\sigma_{r}^{2}+\|G_{BB1,k}^{H}F_{1k}^{H}H_{k}\;V_{RF}V_{BB,k}{\|}_{F}^{2}\sigma_{d}^{2}+\sigma_{r}^{2}\sigma_{d}^{2}}=\frac{\beta_{1k}\beta_{2k}}{1+\beta_{1k}\beta_{2k}}\;,\;$$
where $\beta_{1k}=\frac{\|G_{BB1,k}^{H}F_{1k}^{H}H_{k}\;V_{RF}V_{BB,k}{\|}_{F}^{2}}{\sigma_{r}^{2}}$ and $\beta_{2k}=\frac{\|W_{BB,k}^{H}W_{RF}^{H}G_{k}\;F_{2k}G_{BB2,k}{\|}_{F}^{2}}{\sigma_{d}^{2}}$ are the SNRs at the *k*th relay node and destination, respectively. As$\;\beta_{1k},\;\beta_{2k}>0$, the overall SNR from the source to the destination achieves maximum level when both$\;\beta_{1k}$ and$\;\beta_{2k}$ attain their maximum values. This target can be achieved by decomposing end-to-end SNR (13) into two independent sub-problems. One sub-problem tries to maximize$\;\beta_{1k}$ and the other attempts to maximize$\;\beta_{2k}$.

Let Vk=VRFVBB,k∈ℂNt×Ns and Wk=WRFWBB,k∈ℂNr×Ns be the hybrid beamforming matrices at the source and destination, respectively. Therefore, the compact representation of the received signal at the destination can be written as
(14)yD=WkHGkFkHk Vksk+WkHGkFkzr+WkHzd∈ℂNs×1. 

The power constraint at the *k*th relay node is given as
(15)$$E\left\{\|F_{k}H_{k}\;V_{k}s_{k}+F_{k}z_{r}{\|}_{F}^{2}\right\}\;\leq P_{r},\;$$
where $P_{r}$ is the transmit power at the *k*th relay station. Using (14), the achievable rate, when transmitted data streams pass through the selected *k*th relay node, can be obtained by the following relation
(16)$$C_{k}=\left(\frac{1}{2}\right)log_{2}det\left\{I_{N_{s}}+\left(W_{k}{}^{H}G_{k}F_{k}H_{k}\;V_{k}\right)R_{n}^{-1}{\left(W_{k}{}^{H}G_{k}F_{k}H_{k}\;V_{k}\right)}^{H}\right\},\;$$
where$\;R_{n}=\sigma_{r}^{2}\left(W_{k}{}^{H}G_{k}F_{k}\right){\left(W_{k}{}^{H}G_{k}F_{k}\right)}^{H}+\sigma_{d}^{2}W_{k}{}^{H}W_{k}$ is the equivalent noise covariance matrix. The optimization problem to maximize the achievable rate through the *k*th relay node can be formulated as
$$\underset{\left[V_{RF},\;V_{BB,k},\;F_{1k},\;G_{BB1,k},\;G_{BB2,k}\;,\;F_{2k}\;,\;W_{RF},\;W_{BB,k}\right],\forall k\in \left\{1,\ldots ,K\right\}}{\max}\beta_{k}$$
$$s.t.\;\|V_{RF}V_{BB,k}{\|}_{F}^{2}\leq P_{s}\;,E\left\{\|F_{k}H_{k}\;V_{k}s_{k}+F_{k}z_{r}{\|}_{F}^{2}\right\}\leq P_{r}$$
(17)$$\left|V_{RF}\left(x,y\right)\right|=\frac{1}{\sqrt{N_{t}}}\;,\left|F_{1k}\left(x,y\right)\right|=\frac{1}{\sqrt{N_{rel}}},\left|F_{2k}\left(x,y\right)\right|=\frac{1}{\sqrt{N_{rel}}},\;\left|W_{RF}\left(x,y\right)\right|=\frac{1}{\sqrt{N_{r}}},\forall \;x,y,\;$$
where $P_{s}$ denotes the transmit power at the source. 

Generally, the interest lies in joint optimization of the problem in (17) for deriving the required analog RF and digital baseband precoders/combiners. In the presence of several element-wise constant amplitude constraints, joint optimization of (17) is quite challenging for finding the global optimal solution. These element-wise constant modulus constraints and several matrix variables make the problem in (17) non-convex, and hence, mathematically intractable. In this perspective, a suboptimal solution is expected by following a de-coupled approach, as did in [45], where phase-only precoding/combining and baseband precoders/combiners are designed separately to maximize the achievable rate. Therefore, the solution to the problem formulated in (17) depends on the suitable choice of the following optimization variables
$$V_{RF},V_{BB,k},\;F_{1k},\;G_{BB1,k},\;G_{BB1,k},\;F_{2k},\;W_{BB,k},\;W_{RF}.$$

## 3. Proposed Analog RF Beamforming

This section deals with the derivation of phase-only precoder at source, the analog RF beamforming matrices at relay nodes and the RF combiner at the destination.

### 3.1. Source Analog Precoder

Multi-linear singular value decomposition (SVD) can be used to derive the common RF beamformer at the source that enables transmission of data streams through any selected relay node. The tensor representation of channel matrices from source to relay nodes is given as
(18)Ht=[H1⌴3…⌴3HK]∈ℂNrel×Nt×K, 
where H1, H2,…HK are the channel matrices from the source to K relay nodes. Multi-linear SVD of Ht is expressed as
(19)Ht=S×1X(1)×2X(2)×3X(3), 
where S∈ℂNrel×Nt×K is a core tensor, and X(1)∈ℂNrel×Nrel, X(2)∈ℂNt×Nt and X(3)∈ℂK×K are unitary matrices. It is worth highlighting that X(2)∈ℂNt×Nt contains orthonormal bases for the column space spanned by [H1T…HKT]∈ℂNt×KNrel. Therefore, the full-complexity beamforming matrix Fopt∈ℂNt×NtRF consisting of first NtRF columns of X(2) can facilitate in designing the common analog RF precoder at the source. Hence,
(20)Fopt=X(2)(:,1:NtRF)∈ℂNt×NtRF. 

The unconstrained beamforming matrix $F_{opt}$ in (20) is not favorable due to the absence of element-wise constant amplitude constraint. To resolve this issue, the optimization problem that minimizes the reconstruction loss between $F_{opt}$ and $V_{RF}$, where $V_{RF}$ is the common analog RF beamformer at the source with the element-wise constant modulus constraint, is formulated. In an attempt to minimize the reconstruction loss between the above-mentioned beamforming matrices, the solution of the following error minimization problem leads to the required phase-shift values for designing the source analog beamformer $V_{RF}$. Therefore,
$$\underset{V_{RF}}{\min}\;\|F_{opt}-V_{RF}{\|}_{F}^{2}=tr\left\{\left(F_{opt}-V_{RF}\right){\left(F_{opt}-V_{RF}\right)}^{H}\right\}$$
(21)$$s.t.\;\|V_{RF}V_{BB,k}{\|}_{F}^{2}\leq P_{s},\;\left|V_{RF}\left(x,y\right)\right|=\frac{1}{\sqrt{N_{t}}},\;\forall \;x,y,\;$$
where $P_{s}$ is the transmitted power at the source. From the above formulation, it is possible to decompose the cost function in (21) into its components as
$$f=\overset{N_{t}^{RF}}{\underset{i=1}{\sum }}\left(F_{opt}\left(:,i\right)-V_{RF}\left(:,i\right)\right){\left(F_{opt}\left(:,i\right)-V_{RF}\left(:,i\right)\right)}^{H}$$
(22)$$=f_{1}+\ldots +f_{N_{t}^{RF}},\;$$
where $\;F_{opt}\left(:,i\right)$ and $V_{RF}\left(:,i\right)$ represent the *i*th column of these matrices. Therefore, the component functions in (22) can be written as
(23)$$\begin{array}{c} f_{1}=\left(F_{opt}\left(:,1\right)-V_{RF}\left(:,1\right)\right){\left(F_{opt}\left(:,1\right)-V_{RF}\left(:,1\right)\right)}^{H}\\ \vdots \\ f_{N_{t}^{RF}}=\left(F_{opt}\left(:,N_{t}^{RF}\right)-V_{RF}\left(:,N_{t}^{RF}\right)\right){\left(F_{opt}\left(:,N_{t}^{RF}\right)-V_{RF}\left(:,N_{t}^{RF}\right)\right)}^{H} \end{array}\;,\;$$

It is obvious from (22) that phase-shift values for designing the beamforming vectors in$\;V_{RF}$ can be obtained by minimizing each component function independently. The general representation of $\;F_{opt}\left(:,i\right)$ and $V_{RF}\left(:,i\right)$, for evaluating the required phase-shift values is given as
(24)$$F_{opt}\left(:,i\right)=\left[\begin{array}{c} \begin{array}{c} r_{1}e^{j\theta_{1i}}\\ r_{2}e^{j\theta_{2i}}\\ \vdots \end{array}\\ r_{N_{t}}e^{j\theta_{N_{t}i}} \end{array}\right],\;V_{RF}\left(:,i\right)=\left[\begin{array}{c} \begin{array}{c} e^{jX_{1i}}\\ e^{jX_{2i}}\\ \vdots \end{array}\\ e^{jX_{N_{t}i}} \end{array}\right].\;$$

Algebraic manipulations after substituting$\;F_{opt}\left(:,i\right)$ and$\;V_{RF}\left(:,i\right)$ in$\;f_{i},\;i\in \left\{1,2,\ldots ,N_{t}^{RF}\right\}$ transforms it into another useful form that helps in finding the phase-shift values that lead to the function at its local minimum. The final simplified form of$\;f_{i}$ can be expressed as
(25)$$f_{i}=\overset{N_{t}}{\underset{m=1}{\sum }}\left\{1+r_{mi}^{2}-2r_{mi}\cos\left(X_{mi}-\theta_{mi}\right)\right\}.\;\;$$

It is clear from (25) that$\;f_{i}$ attains minimum when$\;X_{mi}=\theta_{mi},\forall i,m$.

### 3.2. Destination Analog RF Combiner

Similar to the common analog precoder at the source for transmit beamforming gain over all the channels Hk, k∈{1,…,K}. There is a definite need to derive the common analog combiner WRF∈ℂNr×NrRF at the destination for the RF receive processing over all the channels from the K relay nodes to the destination, i.e., G1, G2,…,GK. This target can be achieved by minimizing the reconstruction loss of WRF over the set of optimal combiners
{Wopt1∈ℂNr×NrRF, Wopt2∈ℂNr×NrRF,…,WoptK∈ℂNr×NrRF}
corresponding to the aforementioned channels Gk, k∈{1,…,K}. Therefore, a problem is formulated that attempts to minimize the error of a single matrix over several matrices simultaneously. The solution of this problem leads to the design of the required RF combining matrix. The problem formulation is given as
minWRF{‖WRF−Wopt1‖F2+⋯+‖WRF−WoptK‖F2},
(26)s.t. |WRF(x,y)|=1Nr ∀ x,y. 

It is known that$\;{\left\|A\right\|}_{F}^{2}+{\left\|B\right\|}_{F}^{2}\geq \|A+B{\|}_{F}^{2}$, and therefore, the cost function in (26) can be written as
(27)$$\|W_{RF}-W_{opt1}{\|}_{F}^{2}+\cdots +\|W_{RF}-W_{optK}{\|}_{F}^{2}\geq \|W_{RF}-{\left(\frac{W_{opt1}+\cdots +\|W_{optK}}{K}\right)\|}_{F}^{2}.\;$$

Let
(28)$$W_{opt}=\left(\frac{W_{opt1}+\cdots +W_{optK}}{K}\right).\;$$

Using (27) and (28), the problem (26) can be re-translated into the following form
$$\underset{W_{RF}}{\min}\|W_{RF}-W_{opt}{\|}_{F}^{2}$$
(29)$$s.t.\;\left|W_{RF}\left(x,y\right)\right|=\frac{1}{\sqrt{N_{r}}}\;,\forall \;x,y.\;$$

The global optimal solution to the problem (29) i.e., $\|W_{RF}-W_{opt}{\|}_{F}^{2}=0$ is possible only when the element-wise constant amplitude constraint associated with$\;W_{RF}$ is relaxed, but it is not allowed to relax the constant modulus constant while designing $W_{RF}$, and therefore, $\|W_{RF}-W_{opt}{\|}_{F}^{2}\approx 0$ is expected. This condition leads to the sub-optimal solution of (29) as
(30)$$\|W_{RF}-W_{opt}{\|}_{F}^{2}=\|W_{RF}{\|}_{F}^{2}+\|W_{opt}{\|}_{F}^{2}-2tr\left\{ℜ\left[W_{RF}W_{opt}^{H}\right]\right\},\;$$
where ℜ[.] represents the real part of the input entity, whose minimum is achieved when$\;W_{RF}$ has the same element-wise phase as that of$\;W_{opt}$, thus
(31)$$W_{RF}=\left(\frac{1}{\sqrt{N_{r}}}\right)exp\left\{j\;arg\left(W_{opt}\right)\right\},\;$$
where arg(.) denotes the argument operator.

### 3.3. Relay Analog RF Beamforming

It has already been mentioned in Section 2 that end-to-end SNR maximization can be achieved under the following condition
(32)$$\max\left(\beta_{1k}\right)\max\left(\beta_{2k}\right)\Rightarrow \max\left(\beta_{k}\right),\;$$

This condition makes it possible to decompose the original complicated optimization problem into two sub-problems. One sub-problem leads to the solution of relay analog combiner$\;F_{1k}$ and the other facilitates in evaluating the relay analog precoder$\;F_{2k}$.

#### 3.3.1. Relay Analog Combiner

To derive$\;F_{1k}$, a problem is formulated that attempts to maximize SNR at the receiving end of the *k*th relay node. Therefore,
$$g_{k}=\underset{F_{1k},k\in \left\{1,\;2,\ldots ,K\right\}}{\max}\frac{\|F_{1k}^{H}H_{k}\;F_{opt}{\|}_{F}^{2}}{\sigma_{r}^{2}}$$
(33)$$s.t.\;\left|F_{1k}\left(x,y\right)\right|=\frac{1}{\sqrt{N_{rel}}}\;\forall \;x,y.\;$$

It is assumed that channel state information $H_{k}$ is available at the relay node and $F_{opt}$ has already been derived (20), and hence, it is possible to define the composite channel$\;H_{eff,k}=H_{k}\;F_{opt}$ from the source to the *k*th relay node. Therefore, the problem in (33) can be transformed into another useful form as
$$g_{k}=\underset{F_{1k},k\in \left\{1,\;2,\ldots ,K\right\}}{\max}\frac{\|F_{1k}^{H}H_{eff,k}{\|}_{F}^{2}}{\sigma_{r}^{2}}$$
(34)$$s.t.\;\left|F_{1k}\left(x,y\right)\right|=\frac{1}{\sqrt{N_{rel}}},\;\forall \;x,y.\;$$

To tackle the problem in (34), it can be translated into another equivalent form such as
(35)$$g_{k}=\underset{F_{1k}}{\max}\left\{\frac{\|F_{1k}^{H}H_{eff,1}{\|}_{F}^{2}}{\sigma_{r}^{2}},\ldots ,\frac{\|F_{1k}^{H}H_{eff,K}{\|}_{F}^{2}}{\sigma_{r}^{2}}\right\}\;.\;$$

It is known that$\;F_{1k}$ can be derived by maximizing the composite channel $H_{eff,k}$ gain,∀k∈{1,…,K}. Therefore, the optimization problem in (35) further boils down to the gain maximization of$\;H_{eff,k}\;$over a set of different k values such as
(36)$$\underset{k\in \left\{1,\;2,\ldots ,K\right\}}{\max}\left|H_{eff,k}\right|.\;$$

To solve the problem in (36), all gain values need to be sorted and select the value of k over which$\;\left|H_{eff,k}\right|$
attains the maximum value. Phase-shift values are extracted from the 
corresponding $H_{eff,k}$ for designing$\;F_{1k}$. Finally,$\;F_{1k}$ is given by the following relation
(37)$$F_{1k}=\left(\frac{1}{\sqrt{N_{rel}}}\right)e^{j∡H_{eff,k}},\;$$
where $\left(\frac{1}{\sqrt{N_{rel}}}\right)$ is a power normalization constant.

#### 3.3.2. Relay Analog Precoder

To derive$\;F_{2k}$, a similar problem is formulated that attempts to maximize SNR at the destination. Hence,
$$h_{k}=\underset{F_{2k},k\in \left\{1,\;2,\ldots ,K\right\}}{\max}\frac{\|W_{opt}^{H}G_{k}\;F_{2k}{\|}_{F}^{2}}{\sigma_{d}^{2}}$$
(38)$$s.t.\;\left|F_{2k}\left(x,y\right)\right|=\frac{1}{\sqrt{N_{rel}}},\;\forall \;x,y.\;$$

Just like the problem (33), the composite channel $G_{eff,k}=W_{opt}^{H}G_{k}$ from the *k*th relay to the destination can be defined by exploiting the available information. It enables the transformation of (38) into the following form
$$h_{k}=\underset{F_{2k},k\in \left\{1,\;2,\ldots ,K\right\}}{\max}\frac{\|G_{eff,k}F_{2k}{\|}_{F}^{2}}{\sigma_{d}^{2}}$$
(39)$$s.t.\;\left|F_{2k}\left(x,y\right)\right|=\frac{1}{\sqrt{N_{rel}}},\;\forall \;x,y.\;$$

Similarly, the problem (39) can also be transformed into another equivalent form, as that of (33), which helps in deriving the required RF beamformer$\;F_{2k}$. Hence,
(40)$$h_{k}=\underset{F_{2k}}{\max}\left\{\frac{\|G_{eff,1}F_{2k}{\|}_{F}^{2}}{\sigma_{d}^{2}},\ldots ,\frac{\|G_{eff,K}F_{2k}{\|}_{F}^{2}}{\sigma_{d}^{2}}\right\}\;.\;$$

It is possible to obtain $F_{2k}$ by computing the maximum gain of $G_{eff,k}\forall k\in \left\{1,\ldots ,K\right\}$. Therefore, the problem (40) can be transformed into another useful form for designing $F_{2k}$ as
(41)$$h_{k}=\max\left\{\|G_{eff,1}{\|}_{F}^{2},\ldots ,\|G_{eff,K}{\|}_{F}^{2}\right\}.\;$$

The solution to the problem (41) gives the maximum gain of$\;G_{eff,k}$ with index k at which this condition satisfies. Therefore,$\;F_{2k}$ can be obtained by including the element-wise constant modulus constraint as
(42)$$F_{2k}=\left(\frac{1}{\sqrt{N_{rel}}}\right)e^{j∡G_{eff,k}^{H}}.\;$$

#### 3.3.3. Relay Selection Criteria

When multiple relays are deployed between the source and destination for the selection of best available channel to maximize the spectral efficiency then, multi-relay transmission in parallel suffers from severe interference with each other. To address this problem, a relay selection technique can be exploited to leverage spatial diversity gain in practical systems while minimizing the effect of interference at the destination. This problem can be addressed by selecting the best relay for the transmission of data streams while the other relays are kept silent.

Mainly, there are two relay selection schemes. The first scheme is opportunistic relay selection, where a decision is taken by considering the maximum end-to-end SNR. The other relay selection strategy is partial, where some suitable performance metric is chosen (e.g., SNR, SVD etc.) either in the first-hop or the second-hop. It is worth noting that opportunistic relay selection achieves diversity gain equal to the number of relay nodes, whereas the diversity gain for partial relay selection is one [46,47]. Therefore, the proposed hybrid beamforming algorithm adopts opportunistic relay selection to attain better system performance by maximizing SNR from the source to the end user. The index $k^{*}$ of the selected relay can be determined by
$$k^{*}=\arg\underset{k=\left\{1,\ldots ,K\right\}}{\max}\left(\|H_{eff,k}{\|}_{F}+\|G_{eff,k}{\|}_{F}\right)$$
where $H_{eff,k}=H_{k}F_{opt},\;G_{eff,k}=W_{opt}^{H}G_{k}$.

## 4. Digital Baseband Precoding and Combining

After finding the analog RF processing components at all communicating nodes, it is required to derive the corresponding digital baseband precoding/combining matrices. For this purpose, the equivalent baseband channels corresponding to Hk and Gk are defined as
(43)Heqk=F1kHHkVRF∈ℂNrelRF×NtRF, 
(44)Geqk=WRFHGkF2k∈ℂNrRF×NrelRF. 
where Heqk denotes the baseband equivalent channel from the source to the *k*th relay node, and Geqk represents the baseband equivalent channel from the *k*th relay node to the destination. The received signal at the output of destination RF combiner WRF (31) using (43) and (44) can be written as
yD−=WRFHGkF2kGBB,kF1kHHk VRFVBB,ksk+WRFHGkF2kGBB,kF1kHzk+WRFHzd∈ ℂNrRF×1
(45)=GeqkGBB,kHeqkVBB,ksk+n=HVBB,ksk+n, 
where H=GeqkGBB,kHeqk∈ℂNrRF×NtRF is the equivalent baseband channel from the source to the destination, and n=WRFHGkF2kGBB,kF1kHzk+WRFHzd∈ ℂNrRF×1 is the equivalent noise vector at the destination. It is noteworthy that H is a function of combined baseband processing unit GBB,k at the *k*th relay node. The baseband combiner WBB,k at the destination is used to estimate sk, which is given as
(46)sk−=WBB,kHyD−. 
where sk− is the estimated information symbol vector at the destination when data transmission takes place through the *k*th relay node. To enhance the system performance, it is desired to design the baseband processing units that enable end-to-end SNR maximization. This goal can be achieved by formulating a problem based on MSE, which can be expressed as
(47)M=E{sk−−sk2}. 

The minimization of (47) provides the optimal baseband combiner$\;W_{BB,k}^{H}$ [48] as
(48)$$W_{BB,k}^{H}={\left(HV_{BB,k}\right)}^{H}{\left\{\left(HV_{BB,k}\right){\left(HV_{BB,k}\right)}^{H}+R_{N}\right\}}^{-1},$$
where$\;R_{N}=E\left[nn^{H}\right]=\sigma_{r}^{2}\left(G_{eqk}G_{BB,k}\right){\left(G_{eqk}G_{BB,k}\right)}^{H}+\sigma_{d}^{2}I_{N_{r}}$ is the equivalent noise covariance matrix. Using (48), the estimated symbol vector$\;s_{k}^{-}$ (46) can be written as
(49)$$s_{k}^{-}={\left(HV_{BB,k}\right)}^{H}{\left\{\left(HV_{BB,k}\right){\left(HV_{BB,k}\right)}^{H}+R_{N}\right\}}^{-1}y_{D}^{-}.\;$$

Substituting (48) into (47) leads to the following MMSE formulation as
(50)$$M_{min}=tr\left(D\right)=tr\left\{{\left(I_{N_{s}}+E\right)}^{-1}\right\},\;$$
where$\;E=V_{BB.k}^{H}H_{eqk}^{H}G_{BB.k}^{H}G_{eqk}^{H}{\left\{\sigma_{r}^{2}G_{eqk}G_{BB,k}G_{BB.k}^{H}G_{eqk}^{H}+\sigma_{d}^{2}I_{N_{r}}\right\}}^{-1}\times G_{eqk}G_{BB,k}H_{eqk}V_{BB,k}$ and this expression depends on$\;V_{BB,k}$ and$\;G_{BB,k}$. The design objective here is to find these two precoding matrices that would be able to minimize MSE. Therefore, the optimization problem can be formulated as
$$\underset{V_{BB,k},G_{BB,k}}{\min}tr\left(D\right)=\overset{N_{s}}{\underset{n=1}{\sum }}D_{n,n}\;$$
$$s.t.\;D={\left\{I_{N_{s}}+V_{BB.k}^{H}H_{eqk}^{H}G_{BB.k}^{H}G_{eqk}^{H}\left\{\sigma_{r}^{2}G_{eqk}G_{BB,k}G_{BB.k}^{H}G_{eqk}^{H}+\sigma_{d}^{2}I_{N_{r}}\right\}{\;}^{-1}\times G_{eqk}G_{BB,k}H_{eqk}V_{BB,k}\right\}}^{-1},$$
$$\|V_{BB,k}V_{RF}{\|}_{F}^{2}\leq P_{s}\forall k,$$
(51)$$tr\left(E\left[F_{k}y_{k}y_{k}^{H}F_{k}^{H}\right]\right)=tr\left(F_{k}\left\{\left(H_{k}\;V_{RF}V_{BB,k}\right){\left(H_{k}\;V_{RF}V_{BB,k}\right)}^{H}+\sigma_{r}^{2}I_{N_{rel}}\right\}F_{k}^{H}\right)\leq P_{r}\forall k.\;$$

The problem (51) is also non-convex, and hence, mathematically intractable. Therefore, a sub-optimal solution is expected to maximize the spectral efficiency (16). Furthermore, the baseband precoding matrices $V_{BB,k},G_{BB,k}$ must fulfill the transmit power constraint at the source and each relay node. This curtailment leads to the immediate conclusion that the optimal solution is difficult to derive. In addition, the objective function involves a series of matrix inversions and multiplications, and it shows a complex and non-linear function of baseband precoding matrices, which needs to be designed.

Sub-optimal solution of $G_{BB,k}$ can be obtained analytically through MSE diagonalization procedure [48]. Considering this motivation factor, $G_{BB,k}$ can be evaluated by performing diagonalization of the equivalent channels$\;H_{eqk}$, $G_{eqk}$, when observed from the corresponding baseband processing units. It is worth highlighting that SVD, of these two equivalent channels, facilitates in deriving $G_{BB,k},\;V_{BB,k}$ as
(52)$$H_{eqk}=U_{1}S_{1}V_{1}^{H},\;$$
(53)$$G_{eqk}=U_{2}S_{2}V_{2}^{H}.\;$$

It is obvious from (52) and (53) that diagonalization of D in (51) can be obtained by setting GBB,k=V2(:,1:Ns)U1(:,1:Ns)H and VBB,k=V1(:,1:Ns) where U1∈ℂNrelRF×NrelRF, V1∈ℂNtRF×NtRF, V2∈ℂNrelRF×NrelRF are unitary matrices. The proposed algorithm is summarized in Table 1.

### 4.1. Complexity Analysis

Hybrid beamforming algorithm with relay selection mechanism was proposed to enhance the spectral efficiency by deploying multiple relays between source and destination. Multi-linear SVD is applied on the overall channel Ht∈ℂNrel×Nt×K from the source to K relay nodes for finding the common orthonormal bases, and then, the reconstruction loss is minimized to derive the phase-shift values for designing the required source analog beamformer. This process requires computational cost in the order of O(QNtNrelK), where Q is defined as Q:=max{NtNrelK}. The computational complexity in deriving the analog combiner at destination is in the order of matrix addition. Moreover, the size of this matrix depends on the number of antennas Nr and the number of RF-chains NrRF. Therefore, it requires approximately Nr×NrRF operations to evaluate the desired RF combiner. In addition, the major contributing factor in evaluating the analog precoder/combiner at relay node is the matrix multiplication. Hence, Nrel×Nt×NrelRF operations need to be performed while deriving the relay analog beamformer, and NrRF×Nr×Nrel operations are required for designing the relay analog combiner. Digital baseband processing components at communicating nodes require complexity in the order of O(Ns3). Finally, the overall computational cost of the proposed hybrid beamforming scheme can be approximated by the following expression
(54)C≈(KNt2Nrel+NrNrRF+NrelNtNrelRF+NrRFNrNrel)+Ns3, 

## 5. Computer Simulations

In this section, computer simulations are conducted to evaluate the performance of the proposed algorithm for hybrid beamforming and relay selection mechanism in a single-user multi-relay MIMO system. In addition to the fully digital precoding, the performance of the proposed scheme is compared with conventional relay selection techniques (e.g., best-harmonic-mean (BHM) and SVD-based approaches). Sparse mm-Wave channel and rich scattering environment are considered by changing system parameters to generate simulation results. It is assumed that relay nodes are spatially distributed, and hence, there is no interference among the channels in the first hop. To capture the mathematical structure of mm-Wave propagation environment, a narrowband clustered channel is adopted based on the extended Saleh-Valenzuela model given in [23]. The mathematical model of mm-Wave channel with uniform planer (UPA) can be expressed as
(55)$$H_{mm-Wave}=\sqrt{\frac{N_{t}N_{r}}{N_{cl}N_{ray}}}\overset{N_{cl}-1}{\underset{i=0}{\sum }}\overset{N_{ray}}{\underset{l=1}{\sum }}\alpha_{il}\wedge_{t}\left(\varphi_{il}^{t},\theta_{il}^{t}\right)\wedge_{r}\left(\varphi_{il}^{r},\theta_{il}^{r}\right)a_{r}\left(\varphi_{il}^{r},\theta_{il}^{r}\right)\;\;a_{t}{\left(\varphi_{il}^{t},\theta_{il}^{t}\right)}^{H},$$
where $\alpha_{il}$ represents complex gain of the$\;l^{th}$ propagation path in the *i*th scattering cluster with $\alpha_{il}\sim$ $CN\left(0,\sigma_{\alpha ,i}^{2}\right)$, and $\left(\varphi_{il}^{t},\varphi_{il}^{t}\right)$ and $\left(\theta_{il}^{r},\theta_{il}^{r}\right)$ are its angles of departure AoDs and angles of arrival AoAs in azimuth and elevation, respectively. The vectors$\;a_{t}\left(\varphi_{il}^{t},\theta_{il}^{t}\right)$ and $a_{r}\left(\varphi_{il}^{r},\theta_{il}^{r}\right)$ are the normalized transmit and receive array response vectors, and the functions$\;\wedge_{t}\left(\varphi_{il}^{t},\theta_{il}^{t}\right)$ and$\;\wedge_{r}\left(\varphi_{il}^{r},\theta_{il}^{r}\right)$ specify transmit and receive antenna gains, respectively. Moreover, $N_{t},\;N_{r},\;N_{cl}$, and $N_{ray}$ denote the number of transmit antennas, receive antennas, number of clusters, and the number of rays per cluster, respectively.

From the perspective of conventional relay selection strategies in MIMO communication networks, the optimal relay can be selected using the BHM method. The mathematical formulation of this process is given as
(56)$$h_{BHM}=\underset{k\in \left\{1,\ldots ,K\right\}}{\max}\left\{\frac{1}{\left({\left|H_{k}\right|}^{-2}+{\left|G_{k}\right|}^{-2}\right)}\right\},\;$$
where $\left|H_{k}\right|$ denotes the channel gain from source to the *k*th relay node, and$\;\left|G_{k}\right|$ represents the channel gain from the *k*th relay node to the destination. Another method that selects the best relay, for spectral efficiency maximization, is based on the sum of singular values of the channel matrix between the source and relay node. Mathematical formulation of this scheme can be expressed as
$$H_{k}=U_{1k}S_{1k}V_{1k}^{H}$$
(57)$$SR=\underset{k\in \left\{1,\ldots ,K\right\}}{\max}\left\{tr\left(S_{1k}\right)\right\}\;,\;$$
where SR stands for the best selected relay that attains a maximum sum of singular values over all the channels from the source to relay nodes. It is worthwhile to mention here that relay selection, in multi-relay MIMO networks, is a non-convex problem regardless of beamforming strategy (fully digital/hybrid). From this perspective, there is no guarantee for the global optimal solution even with fully digital precoding. This feature can be exploited to obtain the full-complexity solution from the corresponding hybrid transceiver design. Following this approach, a hybrid beamforming is designed by maximizing end-to-end SNR with the associated essential constraints. There exists a high probability of obtaining near-optimal solution for a carefully derived hybrid precoding scheme under given constraints. As the primary focus of the proposed algorithm is on the hybrid transceiver design with relay selection mechanism, and therefore, the performance of fully digital beamforming can be assumed by relaxing the constant amplitude constraints associated with phase-only precoding/combining at all communicating nodes.

### 5.1. Spectral Efficiency Performance with UPA and mm-Wave Channel

Figure 2 illustrates the spectral efficiency performance of the proposed hybrid precoding technique when mm-Wave propagation environment is considered to generate simulation results. Furthermore, UPA is taken into account for antenna deployment, and mm-Wave channel parameters are initialized as $N_{ray}=10,\;N_{cl}=4,\;\sigma_{\alpha ,i}^{2}=1\;\forall i$, and $AS=10^{{}^{\circ}}$ to generate the channel matrix according to (55). These channel parameters describe the number of rays/cluster with Laplacian distributed azimuth and elevation angles of arrival and departure, number of clusters, average power of each cluster, and standard deviation, respectively. Moreover, the angle spread for transmitter sector is supposed to be $60^{{}^{\circ}}$ and$\;20^{{}^{\circ}}$ in the azimuth and elevation domains, respectively [23]. Simulation results are generated by changing the number of antennas and transmitted data streams simultaneously to show the effectiveness of the proposed method. Figure 2a plots the spectral efficiency achieved by the proposed algorithm when the number of antennas at communicating nodes and data streams are set as $N_{t}=N_{rel}=100,\;N_{r}=36,$ and $N_{s}=2,\;3,\;4$, respectively. The number of RF chains is equal to the number of transmitted data streams, i.e., $N_{t}^{RF}=N_{rel}^{RF}=N_{r}^{RF}=N_{s}$. It is evident from the obtained results in Figure 2a that a minor performance gap exists between the proposed approach and full-complexity beamforming. However, the performance gap may increase when a considerably large number of data streams are transmitted. Figure 2b shows the performance of the proposed hybrid beamforming by increasing the number of antennas and data streams such that $N_{t}=N_{rel}=144,\;N_{r}=36$, and $N_{s}=3,\;4,\;5$, respectively. It is clear from the obtained results, in Figure 2b, that the proposed algorithm approaches the upper bound through the best selection of the relay node. Figure 2c plots the spectral efficiency by further increasing the number of antennas at communicating nodes and transmitted data streams such that $N_{t}=N_{rel}=256,\;N_{r}=64$, and $N_{s}=5,\;6,\;7$ to show the usefulness of the proposed methodology. Again, there is a small performance gap when comparison is made with fully digital precoding. In conclusion, the proposed algorithm achieves good performance consistently when system parameters are changed over a wide range. Moreover, significantly better performance is obtained in comparison to the BHM and SVD-based relay selection strategies.

### 5.2. Spectral Efficiency Performance in Rich Scattering Environment

Figure 3 plots the spectral efficiency performance of the proposed hybrid beamforming scheme in a rich scattering environment, which is generated according to Rayleigh fading model, where the elements of the channel matrix are independent complex Gaussian numbers with zero mean and unit variance. Simulation results are obtained by changing the number of antennas at communicating nodes and the number of transmitted data streams as well. It is worth highlighting that the number of RF chains is equal to the number of transmitted data streams, while conducting computer simulations. The condition on the number of RF chains indicates the worst-case scenario, as the number of RF chains cannot be less than the number of data streams according to the condition (2) given in Section 2. Figure 3a shows the spectral efficiency achieved by the proposed method when $N_{t}=N_{rel}=N_{r}=36$, and $N_{s}=2,\;4,\;6$. 

The proposed algorithm approaches the upper bound defined by full-complexity precoding, which is clear from the obtained results in Figure 3a. In the next use case, the number of antennas deployed at communicating nodes increases along with the transmitted data streams, in comparison to the previous one, for the evaluation of spectral efficiency under different system configuration parameters. Therefore, the number of antennas and data streams are set as $N_{t}=N_{rel}=64,\;N_{r}=36$, and $N_{s}=4,\;6,\;8$, respectively, to generate simulation results as shown in Figure 3b. It is obvious from the obtained results that the proposed technique achieves a performance close to fully digital beamforming. Figure 3c illustrates the spectral efficiency performance by further increasing the number of antennas and data streams, in comparison to both the previous cases, such that $N_{t}=N_{rel}=100,\;N_{r}=25,$ and $N_{s}=6,\;8,\;10$. Again, there is a minor performance gap between the proposed approach and its fully digital counterpart as depicted in Figure 3c. In summary, the proposed algorithm achieves near optimal performance in a consistent manner under various system parameters when Rayleigh fading channel comes into play. Moreover, the proposed scheme shows a slightly higher performance when compared with conventional relay selection techniques such as BHM and SVD.

### 5.3. The Optimal Relay Selection Mechanism

Figure 4 plots the spectral efficiency as a function of the relay node, where an attempt is made to explain the process of selecting the optimal relay from the given number of active relay stations. To generate simulation results, the number of antennas deployed at communicating nodes is set as $N_{t}=N_{rel}=100,\;N_{r}=25$ for the transmission of $N_{s}=4$ data streams. It is further assumed that 20 active relay nodes are available between the source and destination for the selection of the best relay station. The obtained results in Figure 4 show that spectral efficiency varies with the relay node, as each relay station provides a different value of end-to-end SNR. Moreover, the variation in spectral efficiency depends on the condition of the relay-assisted transmission path from the source to the destination. It is important to mention that the proposed relay selection mechanism considers end-to-end SNR maximization, which is characterized by the suitable choice of analog RF beamforming matrix and the corresponding digital baseband processing component at the source and each relay node and destination, whenever the hybrid transceiver design is concerned. Since SNR is directly related to capacity expression, a relay selection strategy using end-to-end SNR maximization leads to the maximum achievable rate. It is obvious from Figure 4 that the optimal relay is the one that provides the best channel for the transmission of a signal from source to destination.

### 5.4. Impact of Data Transmission Paths on Spectral Efficiency

Figure 5 illustrates the impact of increasing the number of data transmission paths L on spectral efficiency performance. The number of antennas at the source, each relay node and destination are set as $N_{t}=64,\;N_{rel}=64,$ and $N_{r}=16$, respectively. Just like the previous simulation results, it is assumed that the number of RF chains is equal to the number of transmitted data streams, i.e., $N_{t}^{RF}=N_{rel}^{RF}=N_{r}^{RF}=N_{s}=4$. In addition, the mm-Wave channel matrix is generated with ULA [49] at L=[5, 10, 20, 30], where each transmission path follows uniformly distributed AoA and AoD in the interval [−π/2, π/2]. The spectral efficiency is evaluated at the aforementioned values of L by keeping all other parameters constant to visualize the effect of this change. It is evident from simulation results in Figure 5 that the spectral efficiency decreases by increasing the value of L. However, the performance of the proposed algorithm is close to its fully digital counterpart irrespective of the number of data transmission paths.

### 5.5. Performance Evaluation with Other Relay-Assisted Hybrid Beamforming Algorithms

Figure 6 demonstrates the transmission rate of the presented hybrid beamforming scheme with the optimal relay selection criterion. The system parameters are set as $N_{t}=N_{rel}=100,\;N_{r}=25$, and $N_{s}=3,\;4,\;5$ to obtain simulation results. Moreover, a sparse mm-Wave propagation environment with UPA is considered for data transmission, where channel parameters are initialized as $N_{ray}=10,\;N_{cl}=4,\;\sigma_{\alpha ,i}^{2}=1\;\forall i$, and $AS=10^{{}^{\circ}}$ to generate the channel matrix according to (55). It is obvious from the obtained results that the proposed method achieves higher performance when compared with the algorithms proposed in [34,36].

Figure 7 compares the performance of the proposed scheme with the relay-assisted hybrid beamforming algorithm in [35]. It is worth mentioning that this technique uses analog beamforming at relay nodes, while the hybrid precoder and combiner are designed at the source and destination, respectively. Due to analog beamforming, each relay node supports only one data stream, i.e., $N_{s}=1$. Therefore, computer simulations are conducted in Figure 7 by changing the number of antennas at communicating nodes only. It is obvious from the obtained results that the proposed approach achieves near-optimal performance when compared with full-complexity precoding and outperforms the algorithm presented in [35].

Table 2 summarizes the performance comparison among between the proposed algorithm and the other relay-assisted hybrid beamforming techniques.

## 6. Conclusions

This paper proposes a hybrid beamforming and relay selection scheme in a non-regenerative multi-relay MIMO network for capacity maximization. The constant modulus constraint associated with the RF processing component makes the problem non-convex, and hence, mathematically intractable. Therefore, it is quite challenging to find the global optimal solution. To address this problem, the expression for end-to-end SNR is derived by converting the composite channel into the equivalent SISO channels. Then, the original complicated problem, based on SNR maximization, is transformed into two independent sub-problems. Furthermore, each sub-problem is tackled by following the decoupled approach to reduce the complexity for deriving the hybrid processing components at different communicating nodes. In particular, the source analog beamformer and relay RF combiner are obtained by maximizing SNR at the receiving end of the relay station. While the phase-only precoder at the relay node and the destination analog combiner are derived by maximizing SNR at the destination. Finally, digital baseband processing components are designed to minimize interference among different sub-channels. Computer simulations were conducted by changing system parameters and propagation environments (i.e., Rayleigh fading channel and mm-Wave channel) to show the effectiveness of the proposed methodology. It is obvious from the obtained results that the proposed algorithm achieves performance close to its fully digital counterpart, and significantly better than conventional relay selection techniques such as BHM and SVD. In addition, the proposed scheme outperforms when compared with several relay-based hybrid beamforming techniques. The extension of this work considering frequency-selective channels is a promising future direction.

## Figures and Tables

**Figure 1 sensors-23-02079-f001:**
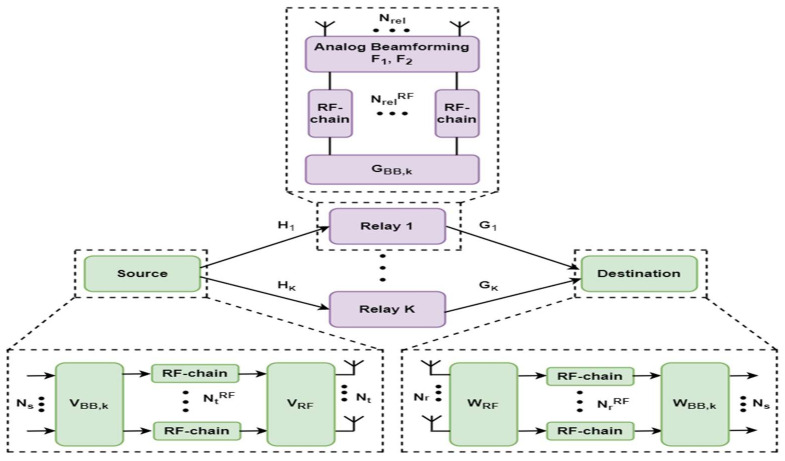
Block diagram of the proposed hybrid beamforming and relay selection for a two-hop multi-relay network.

**Figure 2 sensors-23-02079-f002:**
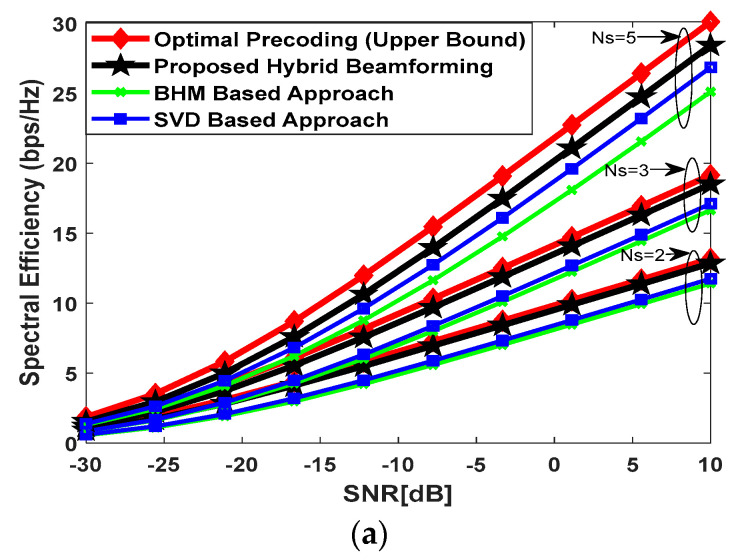
(**a**) Spectral efficiency achieved by different relay selection techniques when$\;N_{t}=N_{rel}=100,\;N_{r}=36,\;N_{s}=2,\;3,\;4$ with UPA at all communicating nodes. Mm-Wave channel matrix is generated by assuming$\;N_{cl}=4,\;N_{ray}=10$ and $10^{{}^{\circ}}$ spread angle. The number of RF-chains is equal to the number of data streams. It is assumed that 10 relay nodes are available between the source and the destination for the selection of the best relay; (**b**) spectral efficiency achieved by different relay selection techniques when$\;N_{t}=N_{rel}=144,\;N_{r}=36,\;N_{s}=3,\;4,\;5$ with UPA at all communicating nodes. Mm-Wave channel matrix is generated by assuming$\;N_{cl}=4,\;N_{ray}=10$ and a $10^{{}^{\circ}}$ spread angle. It is assumed that the number of RF-chains is equal to the number of data streams, and 10 relay nodes are present between the source and the destination for the selection of the best relay; (**c**) spectral efficiency achieved by different relay selection techniques when$\;N_{t}=N_{rel}=256,\;N_{r}=64,\;N_{s}=5,\;6,\;7$ with UPA at all communicating nodes. Mm-Wave channel matrix is generated by assuming$\;N_{cl}=4,\;N_{ray}=10,$ and a $10^{{}^{\circ}}$ spread angle. It is assumed that the number of RF-chains is equal to the number of data streams, and 10 relay nodes are present between the source and the destination for the selection of the best relay.

**Figure 3 sensors-23-02079-f003:**
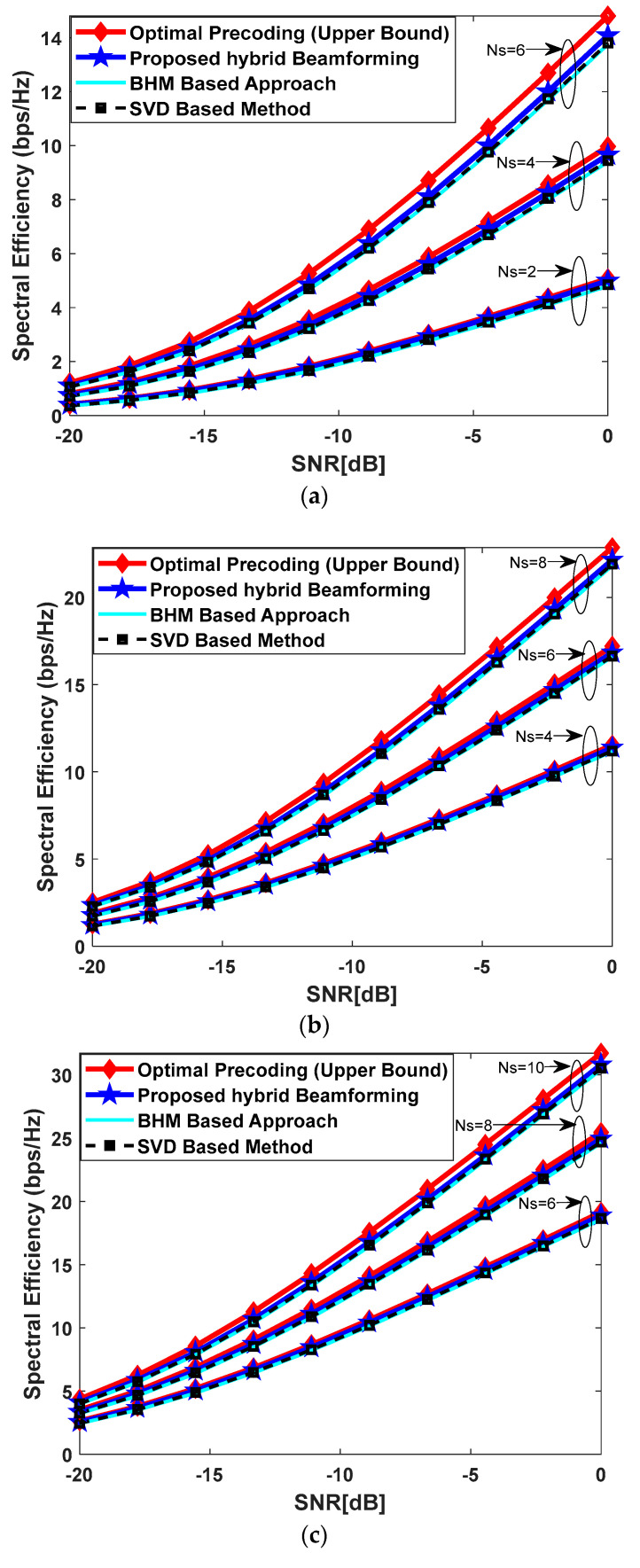
(**a**) Spectral efficiency vs. SNR when$\;N_{t}=N_{r}=N_{rel}=36,\;N_{s}=2,\;4,\;6$ with Rayleigh fading channel and the number of RF-chains is equal to the number of data streams$\;N_{t}^{RF}=N_{r}^{RF}=N_{rel}^{RF}=N_{s}$. There are 10 relay nodes between the source and the destination for the selection of the best relay; (**b**) spectral efficiency vs. SNR when$\;N_{t}=N_{rel}=64,\;N_{r}=36,\;N_{s}=4,\;6,\;8$ with Rayleigh fading channel and the number of RF-chains is equal to the number of data streams$\;N_{t}^{RF}=N_{r}^{RF}=N_{rel}^{RF}=N_{s}$. There are 10 relay nodes between the source and the destination for the selection of the best relay; (**c**) spectral efficiency vs. SNR when$\;N_{t}=N_{rel}=100,\;N_{r}=25,\;N_{s}=6,\;8,\;10$ with Rayleigh fading channel and the number of RF-chains is equal to the number of data streams$\;N_{t}^{RF}=N_{r}^{RF}=N_{rel}^{RF}=N_{s}$. There are 10 relay nodes between the source and the destination for the selection of the best relay.

**Figure 4 sensors-23-02079-f004:**
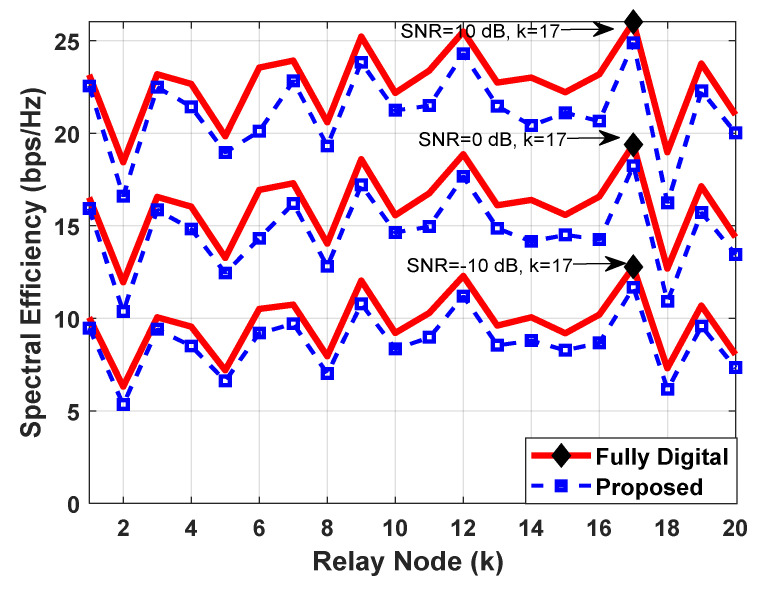
Spectral efficiency vs. relay nodes in a mm-Wave propagation environment with UPA such that $N_{cl}=4,\;N_{ray}=10$ and $10^{{}^{\circ}}$ spread angle. System parameters are set as $N_{t}=N_{rel}=100,\;N_{r}=25,\;N_{s}=4$ with the assumption that the number of RF-chains is equal to the number of data streams$\;N_{t}^{RF}=N_{r}^{RF}=N_{rel}^{RF}=N_{s}$. There are 20 active relay nodes between the source and the destination for the selection of the best relay.

**Figure 5 sensors-23-02079-f005:**
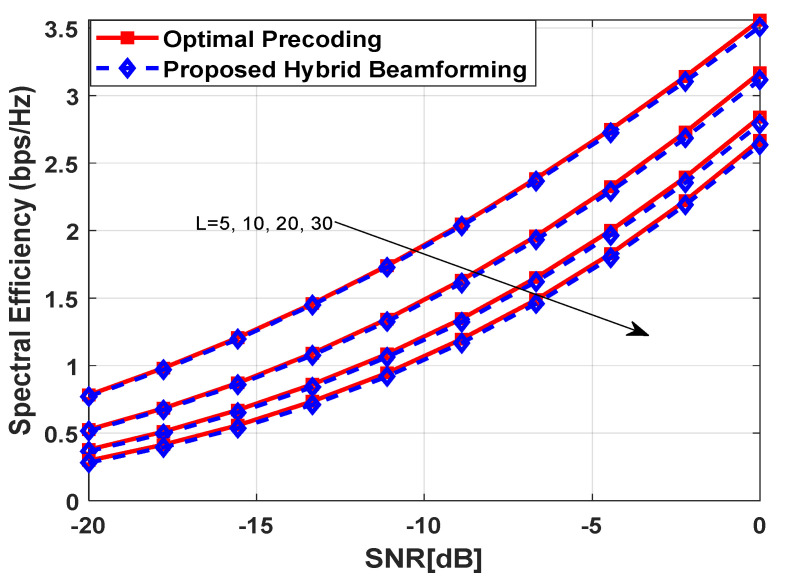
Spectral Efficiency vs. SNR at L=[5, 10, 20, 30] when$\;N_{t}=N_{rel}=64,\;N_{r}=16,\;N_{s}=4$ with ULA and$\;N_{t}^{RF}=N_{r}^{RF}=N_{rel}^{RF}=N_{s}$.

**Figure 6 sensors-23-02079-f006:**
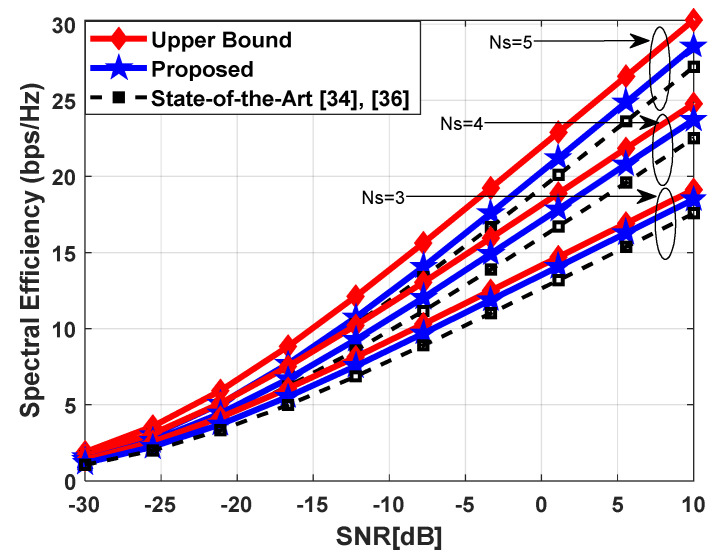
Spectral efficiency vs. SNR in a mm-Wave propagation environment with UPA such that $N_{cl}=4,\;N_{ray}=10$ and $10^{{}^{\circ}}$ spread angle. System parameters are set as $N_{t}=N_{rel}=100,\;N_{r}=25,\;N_{s}=3,\;4,\;5$.

**Figure 7 sensors-23-02079-f007:**
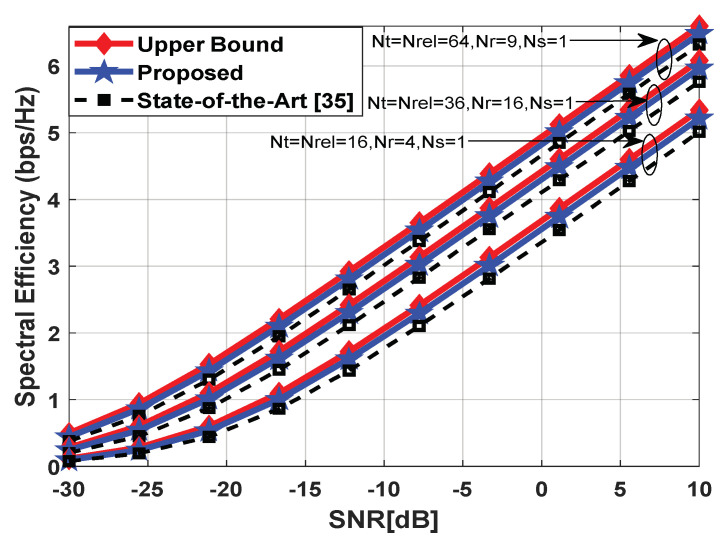
Spectral efficiency vs. SNR in a mm-Wave propagation environment with UPA such that $N_{cl}=4,\;N_{ray}=10,$ and $a\;10^{{}^{\circ}}$ spread angle. The number of antennas is set as $\left(N_{t}=N_{rel}=16,\;N_{r}=4\right),\;\left(N_{t}=N_{rel}=36,\;N_{r}=16\right),\;\left(N_{t}=N_{rel}=64,\;N_{r}=9\right),\;N_{s}=1$.

**Table 1 sensors-23-02079-t001:** Summary of the proposed algorithm.

**Algorithm:** Hybrid transceiver and relay selection in a single-user multi-relay MIMO systems based on end-to-end SNR maximization
1. **Initialization:** Let $\Phi_{1}=\left\{1,2,\ldots ,K\right\},\;I=\left\{1,\ldots N_{t}\right\},\;J=\left\{1,\ldots ,N_{t}^{RF}\right\}$
2. **Input:** Channel Matrix $H_{k},\;G_{k},\;k\in \Phi_{1}$
** First stage: Analog RF beamforming at communicating nodes**
(a) **Source analog beamformer** VRF
3. Overall channel matrix Ht from source to relay nodes is given by (18) Ht=[H1⌴3…⌴3HK]∈ℂNrel×Nt×K
4. Perform multi-linear SVD on Ht (19) Ht=S×1X(1)×2X(2)×3X(3)
5. The un-constrained common beamformer at source can be derived as (20) Fpot=X(2)(:,1:NtRF)∈ℂNt×NtRF
6. The solution of the optimization problem in (21) leads to the required analog RF beamformer VRF=(1Nt)exp(j∡Fpot)
(b) **Relay Analog Beamforming** (F1k, F2k)
7. Define composite channels Hint,k,Gint,k such as Hint,k=HkFopt, Gint,k=WoptHGk,k∈Φ1
8. **for** each relay node k, k=1,2,…,K **do**
9. $S^{*}=max\|H_{int,k}{\|}_{F}$
10. $Z^{*}=max\|G_{int,k}{\|}_{F}$
11. **end for**
12. $F_{1k}=\left(\frac{1}{\sqrt{N_{rel}}}\right)e^{j\;arg\left(S^{*}\right)},\;F_{2k}=\left(\frac{1}{\sqrt{N_{rel}}}\right)e^{j\;arg\left(Z^{*}\right)}$
**Relay Selection:**
13. $k^{*}=\max\left(\underset{\;k\in \Phi_{1}}{\underbrace{\|H_{int,k}{\|}_{F}+\|G_{int,k}{\|}_{F}}}\right)$
(c) **Analog RF combiner** $W_{RF}$ **at destination**
14. $W_{opt}=\frac{1}{K}\left(\sum_{k=1}^{K}W_{opt,k}\right)$
15. Phase-shift values are obtained for designing $W_{RF}$ by solving the optimization problem (29)
16. $W_{RF}=\left(\frac{1}{\sqrt{N_{r}}}\right)e^{j∡\left[W_{opt\left(q,m\right)}\right]}\forall q\in \left\{1,\ldots ,N_{r}\right\},m\in \left\{1,\ldots ,N_{r}^{RF}\right\}$
**Second stage: Digital baseband processing at communicating nodes**
17. Calculate the equivalent baseband channels corresponding to $H_{k},\;G_{k}$ such as $H_{eqk}=F_{1k}^{H}H_{k}V_{RF},\;\;G_{eqk}=W_{RF}^{H}G_{k}F_{2k}$
18. Optimal baseband combiner $W_{BB,k}$, when *k*th relay node is selected for data transmission, can be evaluated by using the expression (48)
19. ***V**_BB,K_* and ***G**_BB,K_* are derived by solving the optimization problem (51) based on MSE diagonalization

**Table 2 sensors-23-02079-t002:** Performance comparison with other algorithms.

Algorithms	Number of Antennas	$N_{s}$	Spectral Efficiency (%)
[34], [36]	$N_{t}=N_{rel}=100,N_{r}=25$	5	90
Proposed	$N_{t}=N_{rel}=100,N_{r}=25$	5	94.5
[34], [36]	$N_{t}=N_{rel}=100,N_{r}=25$	4	91
Proposed	$N_{t}=N_{rel}=100,N_{r}=25$	4	96
[35]	$N_{t}=N_{rel}=16,N_{r}=4$	1	92.5
Proposed	$N_{t}=N_{rel}=16,N_{r}=4$	1	96.5

## Data Availability

Not applicable.

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
