# Peer review of "Hybrid Beamforming and Relay Selection for End-to-End SNR Maximization in Single-User Multi-Relay MIMO Systems"

_sensors, 2023, doi:10.3390/s23042079_

Round 1

Reviewer 1 Report

This article is very well written with the each small concept with the mathematical approach. Appreciated work.

1) What the variables and/or constants in the related equation represent should be stated after the equation.

2) To prove the novelty of proposed algorithm, make a comparison table with the state of art algorithms for the same approach.

3) Include few more results by increasing the numbers of channels to identify the reliability of proposed algorithm.

4)Conclusion is still needed a few detailed description in order to make it more effective.

5) Simulation results for the relay selection criteria needs graphical representation for better clarity of understanding.

Reviewer 2 Report

The article proposed a hybrid beamforming and relay selection method for capacity maximisation in a non-regenerative multi-relay MIMO network. The topic will interest many researchers. However, we have the following concerns about the paper:

·         I suggest for authors to add numerical results or percentage of improvement at the end of the abstract.

  • Fig 1 is not clear; authors should redraw it with high resolution.
  • Revise the size and spaces of equations and the defined parameters.
  • Authors should provide a summary of results outcomes in a table with the related works (i.e., 34-38). to show improvement.
  • Some of the references in this paper are outdated, some of them are even more than 10 years from now. The authors should update references with the last five years papers.
  • Authors should mention the city and country of all cited conferences and proceedings as in 12, 13, 29, 32 to 34, 38, 46.

Therefore, it can be acceptable if the authors address the comments perfectly.

Round 2

Reviewer 2 Report

The authors have responded to all comments accordingly. I recommend acceptance as submitted